# Living through the heat: How urban children and young people experience and envision healthier cities

Constance Bwire[1]*, Rachel Juel[1], James Milner[2], Gabrielle Bonnet[3], Ana Bonell[4], Shunmay Yeung[3], Harshita Umesh[5], Roberto Picetti[1], Sudheer Kumar Kuppili[2], Jessica Gerard[2], Robert Hughes[1]

1 Department of Population Health, London School of Hygiene & Tropical Medicine, London, United Kingdom, 2 Department of Public Health, Environments and Society, London School of Hygiene & Tropical Medicine, London, United Kingdom, 3 Department of Clinical Research, London School of Hygiene & Tropical Medicine, London, United Kingdom, 4 Medical Research Council Unit, London School of Hygiene & Tropical Medicine, Fajara, The Gambia, 5 Bangalore Medical College and Research Institute, Bangalore, India

* constancebwire@gmail.com

## Abstract

Climate change is driving more frequent and intense heatwaves, posing growing risks to urban populations, particularly children and young people (CYP). This study examined how heatwaves affect the health, well-being, and daily lives of CYP across six rapidly urbanising cities: Accra, Kumasi, Ouagadougou, Port Harcourt, Manila, and Dar es Salaam. We conducted online surveys during both heatwave and non-heatwave periods, collecting 2,269 valid responses. Heatwaves were defined as periods when apparent temperatures exceeded the 90th percentile of historical monthly averages for at least three consecutive days, and, where available, confirmed by national meteorological sources. Non-heatwave periods were defined as days within the same month and city when apparent temperatures were less than or equal to the 10-year average daily mean. Participants, CYP aged 13–29 and parents of children under 18, were recruited via targeted digital advertisements on Meta and Google. Quantitative data were analysed using descriptive statistics, chi-square tests, and logistic regression to assess associations between heatwave exposure and self-reported health symptoms, well-being, and daily disruptions. Thematic analysis of open-ended responses identified community priorities for climate resilience. Heatwaves were associated with higher reports of headaches, low mood, anxiety or stress, not enough food, and increased reliance on family support. Adverse effects were more pronounced among younger and lower-income participants. Participants highlighted five priorities for climate-resilient cities: more green spaces, improved water and sanitation, cleaner environments, stronger health and education services, and greater youth participation in decision-making. The results highlight the growing burden of heat-related health symptoms and daily activity disruptions among CYP

**Data availability statement:** All anonymised datasets, data collection tools, and analysis code are publicly available via the London School of Hygiene & Tropical Medicine (LSHTM) Data Compass repository. The corresponding DataCite DOIs are included in the Supporting Information files. All other relevant materials are also provided as Supporting Information.

**Funding:** This study was funded by an award from the Botnar Foundation (Grant number: 101913EH10) to R.H. The funder's website is: https://www.botnarfoundation.org. C.B. was supported full-time by this grant. All other co-authors received part-time salary support from the same grant. The funders had no role in study design, data collection and analysis, decision to publish, or preparation of the manuscript.

**Competing interests:** The authors have declared that no competing interests exist. All salary support was provided through a research grant from the Botnar Foundation. No authors received funding from commercial entities for this work.

and youth-informed strategies to reduce the unequal impacts of extreme heat in urban areas.

## Introduction

Climate change is one of the most pressing health challenges of this century, with serious and unequal impacts across populations [1–3]. Children and young people (CYP) are particularly vulnerable due to their rapid physical, cognitive, and emotional development, limited capacity to reduce exposure, and dependence on caregivers for protection and care [4,5]. Climate-related hazards, including heatwaves, poor air quality, flooding, and food insecurity, have been linked to adverse health outcomes such as asthma and other respiratory illnesses, malnutrition, diarrheal and vector-borne diseases, and mental health conditions like anxiety, depression, and post-traumatic stress [2,4,6–13]. According to the Lancet Countdown [1]a child born today risks living in a world up to four degrees Celsius warmer than the pre-industrial average, substantially increasing their lifetime exposure to extreme heat [1].

These risks are intensified in urban environments, where most CYP live. More than 30% of the world's four billion city dwellers are under the age of 18 [14], and by 2050 nearly 70% of the global population will live in cities [15]. Cities are central to both the causes and consequences of climate change. High population density, resource consumption, and fossil fuel dependence contribute significantly to greenhouse gas emissions [16]. Simultaneously, the built environment, characterized by concrete surfaces, limited green and blue infrastructure, and dense housing, amplifies vulnerability through the Urban Heat Island (UHI) effect. This is a phenomenon whereby cities experience higher surface and ambient temperatures than nearby rural areas due to heat-retaining materials and limited vegetation [17]. These conditions reduce airflow, elevate humidity, and limit nighttime cooling, thereby intensifying heat stress, which often extends beyond the heatwave periods [12,17–20]. Poor housing, overcrowding, and limited access to cooling further increase vulnerability, particularly for CYP living in informal or poorly serviced areas [19,21,22].

This complex interplay of the climatic, infrastructural features, and socio-environmental conditions intensifies the thermal burden and prolongs heat exposure [19,23]. Prolonged exposure prevents the body from releasing accumulated heat, leading to ongoing physiological strain [19,23]. Studies consistently show declines in both sleep duration and quality during multiday heat events and warm nights [19,24,25]. Such disruptions impair recovery and concentration, while also affecting health and daily activities such as learning, physical exercise, and social interaction [19,23,24].

Recent studies confirm CYP's vulnerability to climate impacts, including preterm birth, low birth weight, malnutrition, diarrhoeal disease, and respiratory illnesses [3,6,8]. At the same time, studies point to persistent gaps in areas such as mental health and congenital disorders, as well as limited attention to CYP's perspectives and capacities to contribute to adaptation strategies [6–8,26]. Additionally, while prior

research has explored how climate change affects CYP, few studies capture their real-time lived experiences during acute events across urban settings [8,27,28].

Most studies on CYP and climate change use cross-sectional or retrospective designs, often drawing on qualitative interviews, participatory workshops, or household surveys to examine perceptions and vulnerabilities. These methods yield valuable, context-specific insights [6,8,19,27–29]. However, they are resource-intensive and require long lead times, which makes them difficult to implement rapidly across international cities during acute climate events [7,8,27]. Our previous work [30] demonstrated the feasibility and value of cross-city digital surveys in capturing CYP's experiences with urban air quality. Building on this, this study employed a rapid-deployment, cross-sectional digital survey to address gaps in climate and CYP research.

The digital method was selected for its ability to safely and quickly engage with CYP within 24 hours of heatwave or non-heatwave verification. This allowed for the collection of real-time data during or shortly after exposure, reducing recall bias and supporting international-city data collection. Heatwave periods were defined using meteorological thresholds and, where available, local government recognition.

To capture experiences across diverse urban contexts, this study aimed to: [1] examine how heatwaves and non-heatwave periods affect the daily lives, health, and wellbeing of CYP across six rapidly urbanising cities; and [2] identify CYP-informed priorities for healthier and more resilient cities. The study focused on Accra (Ghana), Kumasi (Ghana), Ouagadougou (Burkina Faso), Port Harcourt (Nigeria), Manila (The Philippines), and Dar es Salaam (Tanzania), which illustrate a range of climatic zones and urbanisation patterns [31–35]. Despite these differences, they share structural challenges, including high population density, widespread informal settlements, limited infrastructure, and prolonged exposure to elevated indoor and outdoor temperatures [33,36,37]. In some of these cities, the presence of major water bodies further complicates local conditions, providing some cooling benefits while increasing humidity and the risk of vector-borne disease [38].

By comparing CYP experiences during and after heatwaves, this study provides timely evidence on how extreme heat disrupts health and daily life in rapidly urbanising cities. It also demonstrates the value of real-time digital surveys for capturing lived experiences and the need for CYP-sensitive approaches in climate adaptation and urban planning.

## Methods

### Ethics statement

Ethical approval was granted by the London School of Hygiene & Tropical Medicine Research Ethics Committee (Ref: 31044). Informed consent was obtained electronically before participation. Parents or legal guardians answered on behalf of children < 18 and provided consent for them. Young adults aged 18–29 completed the survey themselves and provided their own consent. The survey was anonymous and collected no personal identifiers (including IP addresses). Responses were stored on secure, access-restricted, encrypted systems. The study posed minimal risk.

### Study design and setting

Building on our previous study [30], this research employed a cross-sectional design and an online survey to collect data from CYP living in cities during heatwave and non-heatwave periods. Recruitment was conducted online through targeted advertising on Google and Meta platforms, focusing on six cities: Ouagadougou, Port Harcourt, Kumasi, Accra, Manila, and Dar es Salaam (Fig 1). The six cities included were among those where automated systems detected heatwaves during the study period. These detections were verified, where possible, using national meteorological data, allowing for timely survey deployment and comparative analysis. In Manila and Accra, the heatwave periods were verified as recognized heatwaves by the national meteorological agencies, which issued public alerts and warnings at the time. In the other study cities, no such formal recognition was issued, and we relied solely on the heatwave definition. The online

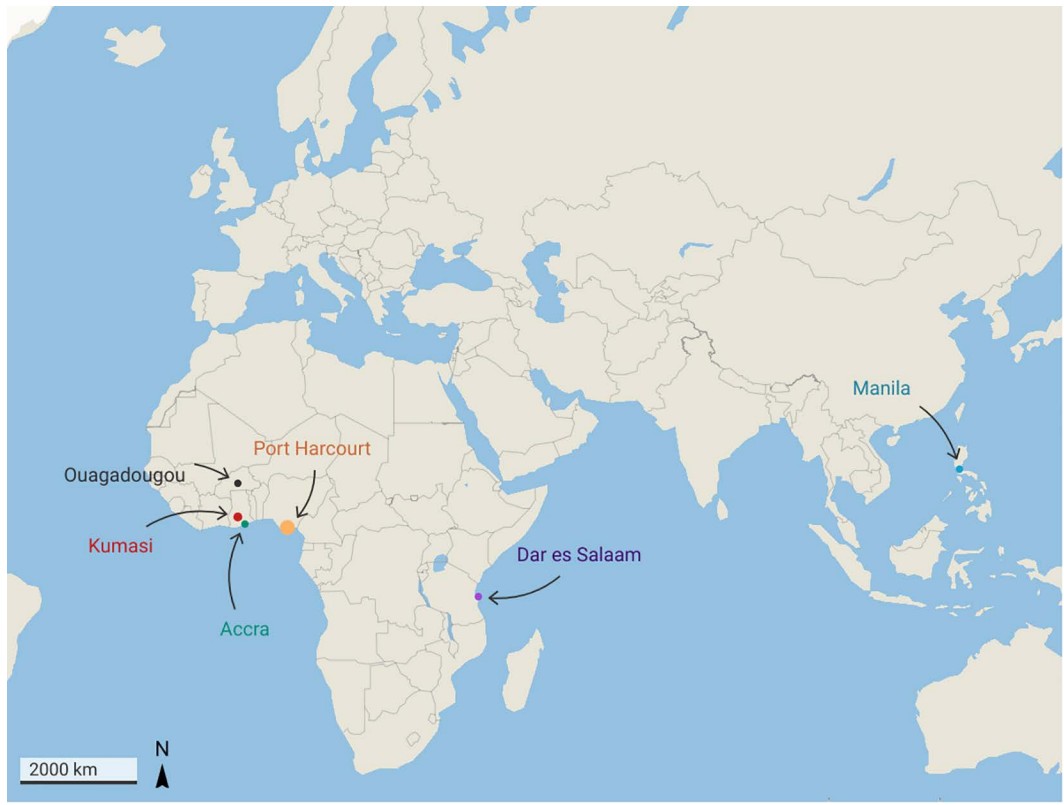

**Fig 1. Location of study cities.** Shows the six urban sites included in the study (Accra, Kumasi, Ouagadougou, Port Harcourt, Manila, and Dar es Salaam), where surveys were conducted during heatwave and non-heatwave periods.

recruitment was selected because it enabled deployment within 24 hours of heatwave verification and ensured standardized, real-time data collection across international cities where longer lead times and local facilitation were not feasible.

This study aimed to capture real-time CYP experiences during heatwave or non-heatwave periods; therefore, characteristics that contribute to UHI effects, such as urban form, density, and surface composition, were not used as city selection criteria. Instead, these characteristics were examined descriptively to contextualize CYP experiences during the verified periods across different cities.

The six cities were selected from a larger pool of 178 focal cities identified by the Children, Cities and Climate (CCC) Action Lab. The CCC Action Lab is a global research initiative led by the London School of Hygiene & Tropical Medicine (LSHTM) focusing on amplifying youth voices and informing equitable climate adaptation in rapidly urbanising settings. These 178 cities were chosen for their youthful populations, fast-paced urbanisation, and relevance to CCC partner networks (see S1 File).

In each city, two survey rounds were conducted: the first during the heatwave, and the second once heatwave conditions had subsided (non-heatwave). This enabled direct comparisons of CYP experiences during heatwave and non-heatwave periods to assess differences in impact.

## Study area

The six study cities, Accra and Kumasi, Ouagadougou, Port Harcourt, Manila, and Dar es Salaam, represent a range of climatic zones across West Africa, East Africa, and Southeast Asia (Fig 1). These cities were selected because they

experienced heatwave events during the study period. In this study, a heatwave was defined as at least three consecutive days when apparent temperatures exceeded the 90th percentile of historical monthly averages (2013–2023), identified through automated monitoring and, where available, confirmed by national meteorological sources.

Accra and Kumasi experience tropical savanna and humid tropical climates, with two rainy seasons (April–June and September–November), high humidity, and hot periods (December–March and July–August) that restrict nighttime cooling [31,32,39]. Ouagadougou, located in the Sahel, has a dry tropical climate marked by long dry seasons (October–May) and a short rainy season (June–September). Extreme daytime heat frequently exceeds 40 °C during March–May [35]. Port Harcourt and Manila both have tropical monsoon climates, with heavy rains from April–October in Port Harcourt and May–October in Manila, followed by shorter dry seasons from November–March and November–April, respectively. In both cities, humidity is high for much of the year, often exceeding 80% during the rainy season but falling to around 65–75% in the drier months. Reduced airflow further heightens thermal discomfort, particularly during the hottest months of March–May [33,40,41]. Dar es Salaam has a humid savanna climate, with two rainy periods (March–May and from October–December) and hot, humid months in between [34]. Across these contexts, average daily temperatures of 22–32 °C combine with humidity and poor nighttime cooling to create persistent thermal discomfort, making heat stress a chronic condition rather than an occasional extreme.

Despite their climatic differences, the cities share common challenges of rapid urbanization, high density, and inadequate infrastructure. Accra (225 km²; 2.5 million; 13,000/km²), Kumasi (254 km²; 3 million; 9,000/km²), and Manila (20,785/km² in 2015) show how migration and population growth have fuelled the spread of informal settlements, overcrowding, and limited basic services [15,18,36,41–43]. Green space has been severely reduced, Accra retains only 15%, Kumasi has lost over 80%, and Ouagadougou's tree cover is below 10% [18,35,42,43]. Blue infrastructure such as lagoons, streams, and reservoirs is often polluted or encroached upon, weakening its cooling role and, in some cases, increasing mosquito-borne disease risks [38]. The dominance of concrete, asphalt, and corrugated metal roofing intensifies UHI effects, with Accra recording up to +3.6 °C above rural surroundings [18,36,44]. In informal settlements, poorly ventilated homes frequently exceed 35 °C indoors [37,45]. Vulnerable groups, including CYP, pregnant women, and older adults, bear the heaviest burden, reporting fatigue, dehydration, sleep disruption, and respiratory strain [37,45,46]. Across all six cities, the interplay of high temperatures, dense urban form, and inadequate services leaves residents, especially vulnerable groups, highly exposed to extreme indoor and outdoor heat. Fig 1 shows the geographic location of the study sites.

## Sampling and participants

A three-stage sampling approach was employed to identify cities and to recruit participants during both heatwave and non-heatwave periods:

**Stage 1: Heatwave identification.** To systematically identify heatwaves across the 178 study cities, we defined a heatwave as a period of at least three consecutive days during which both maximum and minimum daily apparent temperatures exceeded the 90th percentile of historical averages for that calendar month, based on data from the past ten years (2013–2023) [47,48]. Real-time temperature data were retrieved via Open Meteo APIs (https://open-meteo.com/) and integrated with Airtable (https://www.airtable.com/), enabling continuous, automated monitoring across all cities. When cities met the heatwave threshold, Airtable generated alerts to flag potential heatwave and non-heatwave periods. These alerts were then manually reviewed and verified by the research team using publicly available meteorological sources, including national weather service reports.

**Stage 2: Recruitment during heatwave periods.** Once a heatwave event was identified, participants in that city were recruited through paid advertisements (ads) on Meta and Google platforms. These ads were targeted toward two key groups: young people aged 13–29 years, and parents or guardians aged 18 years and older with children under 18. Eligibility criteria included residing in the city currently experiencing a verified heatwave event, having internet access, and

the ability to complete an online survey. Participation was entirely voluntary and anonymous. Each recruitment campaign remained open until either 7 days or until 640 eligible responses were submitted per event, whichever came first (S1 Fig).

**Stage 3: Re-sampling during non-heatwave period.** The survey was repeated in each city following the end of the heatwave. In this study, "non-heatwave periods" conditions referred to the average daily mean apparent temperature for each city during the same month, based on data from the past 10 years [47,49]. As with event identification, the re-sampling process was automated using Airtable, which accessed temperature data through the Open Meteo APIs. Monitoring for non-heatwave event conditions began after the completion of each event survey. Airtable continued to monitor temperatures daily until a suitable non-heatwave event period was identified. S1 Fig shows a detailed workflow.

## Data collection

Data were collected using a short, self-administered online survey hosted on Typeform (https://www.typeform.com/). The survey was promoted through paid advertisements on Meta and Google platforms (S2 File), targeting people in cities experiencing either a verified heatwave event or non-heatwave event conditions. Each survey ran for up to 7 days or until the required number of eligible responses calculated individually per city (S1 Table) was reached.

The survey took around five minutes to complete and included both multiple-choice and open-ended questions (S3 File). It asked participants about their general well-being [50,51], health symptoms, disruptions in daily routines, and how well they felt their city responded to extreme weather. An open-ended question invited suggestions for making cities healthier and more sustainable.

Although the tool was available in twelve languages (S3 File), only the official or most commonly spoken language of each participating city was used. Translations were first generated using ChatGpt generative AI, then reviewed and refined by native speakers. Each version was pre-tested and improved based on feedback from the research project's Youth Advisory Group and early pilot participants. This helped ensure that the survey was clear, accessible, and easy to understand across different cultural and linguistic contexts [51,52].

## Data analysis

Quantitative and qualitative data collected during heatwave and non-heatwave periods were analyzed to explore differences in health, well-being, daily activities, and participant perceptions. Quantitative analyses were conducted using IBM SPSS Statistics Version 30, while qualitative analyses were performed using NVivo Version 15.

**Quantitative analysis.** Descriptive statistics were used to summarize demographic characteristics, health and well-being variables, physical activity length, and daily disruptions. Data were stratified by heatwave event and non-heatwave event conditions. For continuous variables such as age and child's age, independent samples t-tests were used. Categorical indicators such as gender, reported health symptoms, and daily disruptions were analyzed using Pearson's chi-square tests.

To assess the association between heatwaves and non-heatwave periods and health-related outcomes, binary logistic regression models were employed. Ordinal logistic regression was used for outcomes such as self-reported general health and sleep quality, which were measured on a five-point ordinal scale ranging from 1 (Very Bad) to 5 (Very Good). These models included event exposure (heatwave or non-heatwave), demographic covariates (age, gender, income, city), and heatwave conditions as predictors. Binary logistic regression was used to analyze dichotomous outcomes, including the presence or absence of health symptoms (e.g., headaches, anxiety, respiratory issues) and daily activity disruptions (e.g., missed work or school) (S6 and S9 Tables). These models controlled for key demographic and environmental covariates to evaluate the likelihood of adverse outcomes during heatwave compared to non-heatwave periods.

**Qualitative analysis.** Open-ended responses on how cities could be improved were analyzed using a thematic analysis approach in NVivo Version 15. A five-step process guided the analysis: familiarization with the data through iterative reading and keyword searches; initial coding using both inductive and in vivo approaches; theme development by grouping related codes; review and validation using NVivo's coding comparison tool to ensure consistency; and final

Global Public Health

theme definition supported by illustrative quotes. This approach allowed for the identification of common priorities shared amongst participants (S5 File for the detailed Data Analysis Protocol).

**Data management and cleaning.** All survey responses were securely stored on Typeform servers, which are hosted on Amazon Web Services infrastructure. Responses were automatically linked to Google Sheets in real time, allowing the research team to monitor data quality and participation during each heatwave or non-heatwave periods. Final datasets were downloaded and processed on encrypted, password-protected devices.

Confidentiality was maintained throughout the research. No personally identifiable information, IP addresses, or digital tracking data (e.g., cookies) were collected by the research team.

Survey data were checked for completeness, duplication, and formatting consistency. Translation to English was conducted where necessary. Duplicate responses were identified and removed using the "Identify Duplicate Cases" function in SPSS. Missing data were minimal due to the mandatory nature of most survey questions. Responses such as "I prefer not to say" or "I don't know" were retained as valid and analyzed accordingly. Incomplete responses due to disqualification (e.g., age ineligibility) were not included in the final analysis.

## Study limitations

This study has several methodological limitations. Although we used a standardized meteorological definition to identify heatwave periods, not all cities issued public alerts or formally recognized these events, which may have influenced how participants perceived and reported their experiences. The cross-sectional design meant that different individuals participated in the heatwave and non-heatwave surveys, limiting the ability to draw causal inferences. Digital recruitment methods may have excluded CYP without reliable internet access, particularly those in informal settlements, potentially underrepresenting the most heat-vulnerable populations. In addition, digital surveys cannot provide the depth and contextual richness offered by participatory or community-based approaches, which may yield more nuanced insights into the experiences of vulnerable populations. Furthermore, the study did not incorporate urban characteristics such as green or blue infrastructure, surface materials, or housing density, which are known to influence heat exposure. This reflects a broader limitation of survey-based designs, which are not able to fully account for the complexity of urban systems and their interrelated drivers of heat exposure. Finally, sample sizes varied across cities and survey periods, which may affect the comparability and generalizability of findings.

## Results

Across the six cities studied, a total of 2,269 eligible and consenting participants completed the survey during heatwave and non-heatwave periods. Accra received the highest number of heatwave-related responses (486), followed by Manila (362) and Port Harcourt (312). Non-heatwave responses were highest in Dar es Salaam (228), while other cities recorded notably fewer responses (Fig 2).

Variation in participation volume and response quality was evident across cities and event conditions. Manila during the heatwave period recorded the highest survey engagement, with 28,911 views and 820 submissions, indicating strong public responsiveness during a heatwave. In contrast, Dar es Salaam during the non-heatwave period showed the greatest balance of scale and quality, with 828 submissions and 223 eligible responses. Kumasi during the non-heatwave period exhibited the most efficient response pattern, achieving a completion rate of 48.1% and a high proportion of eligible responses relative to total submissions (see Fig 1 and S2 Table). Only a very small number of duplicates were identified, accounting for less than 0.5% of the total.

## Demographics characteristics

Across all six cities (n = 2269), just over half of the participants identified as parents (52.4%), with similar distributions observed during heatwave and non-heatwave periods. Gender representation was mostly males (57.8%), followed by

---

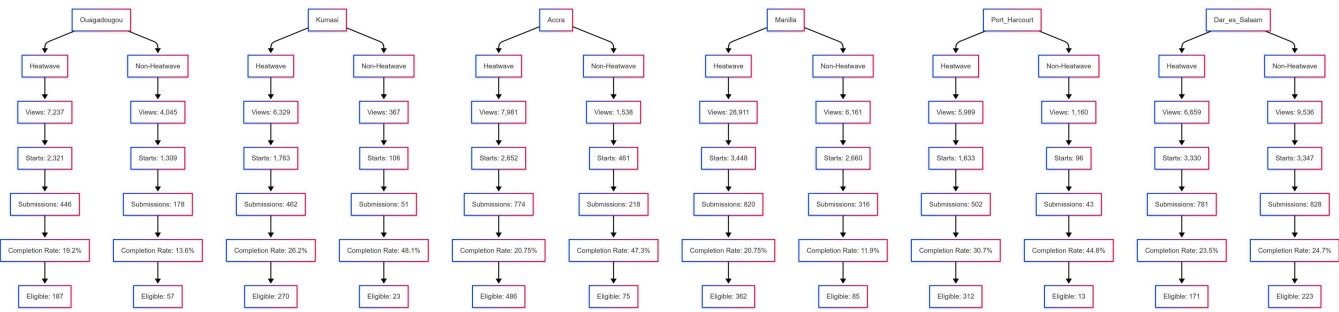

**Fig 2. Survey participation by city and event status.** Shows the number of eligible survey responses collected in each city during heatwave and non-heatwave periods.

females (39.7%), and a small portion (2.6%) who identified as other or preferred not to say. Most participants reported very low incomes (less than $100/ month) or were unsure of their income, highlighting widespread economic uncertainty.

The average age of participants was higher in the heatwave sample (31.5 years) compared to non-heatwave periods (28.9 years), with differences in average ages in Accra (younger participants during heatwaves) and Manila (older participants during heatwaves). The oldest participants overall were in Manila during heatwave periods (41 years). During heatwave conditions, the largest proportion of affected individuals falls within the 20–24 age group (30.8%), followed by 25–29 (23.3%) and 30–39 (16.6%). Conversely, younger individuals [1–14] and older adults (60+) show minimal representation at 0.2% and 0.6%, respectively.

In contrast, during non-heatwave conditions, the age distribution is slightly more balanced. The 20–24 group still represents the highest share at 31.0%, but the 25–29 group rises to 26.0%, with 30–39 decreasing to 14.8%. Notably, the older age group (60+) accounts for a slightly higher percentage than in heatwaves (1.5% vs. 0.6%) (See Fig 3 for age distribution).

Among those who reported on behalf of a child, the average child age was reported as 6.9 years, and remained stable across conditions. Detailed demographic variables across the cities are provided in S3 Table.

**Perceived well-being and health symptoms**

To assess general well-being, participants were asked about how they felt that day, how well they had slept the previous night, and any perceived health symptoms, both physical and mental.

Across all cities (n = 2,269 total event sample; n = 481 for heatwave periods and n = 1,788 for non-heatwave periods) the proportion of participants feeling "Very good" was slightly higher during non-heatwave periods (53.6%) compared to heatwaves (50.7%), while responses of "Good" and "Ok" were similar between the two conditions. Reports of negative emotions ("Bad" or "Very bad") were more frequent during heatwaves (6.2%) than in non-heatwave periods (5.4%) (Fig 4 and S4 Table). Overall, most participants reported positive general well-being under both heatwave and non-heatwave conditions.

Across all cities, sleep quality followed a comparable pattern. The majority of participants reported "Very good" or "Good" sleep regardless of weather conditions, with only small shifts observed between groups. Reports of "Bad" sleep were slightly more common during non-heatwave periods (6.0%) than during heatwaves (5.1%), while "Very bad" sleep was reported more frequently during heatwaves (1.7%) compared to non-heatwave periods (0.8%) (Fig 5 and S4 Table).

Participants across all cities reported higher levels of health symptoms during heatwaves compared to non-heatwave periods (S4 Table). Common symptoms such as headaches (44.1% vs. 37.8%), itchy eyes (30.1% vs. 27.2%), skin irritation (26.8% vs. 22.7%), and low mood (39.0% vs. 32.4%) were more frequently reported during heatwaves. Anxiety

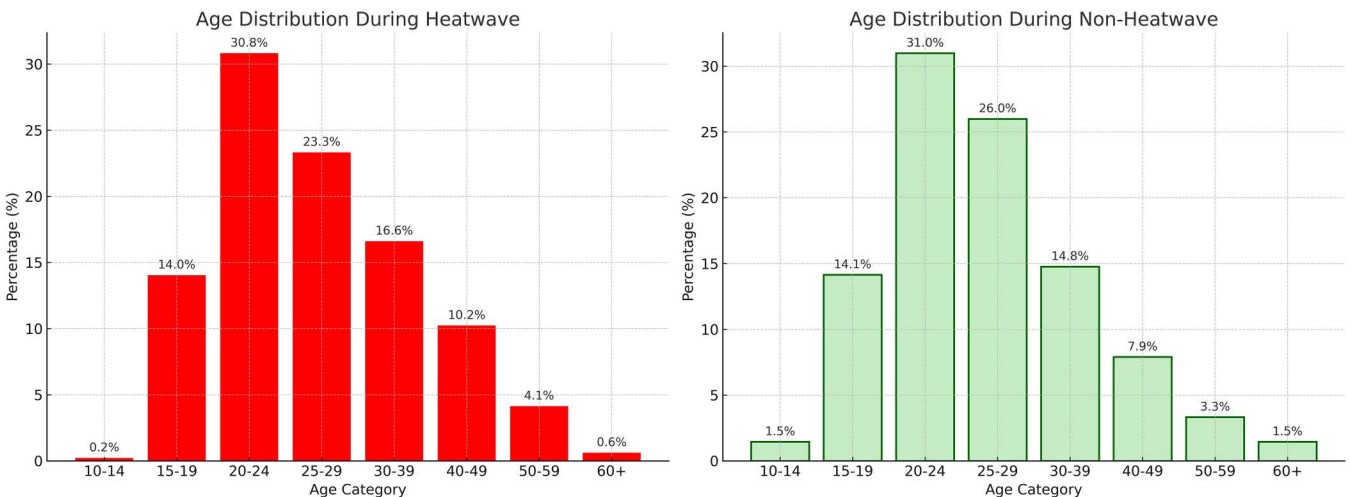

Global Public
Health

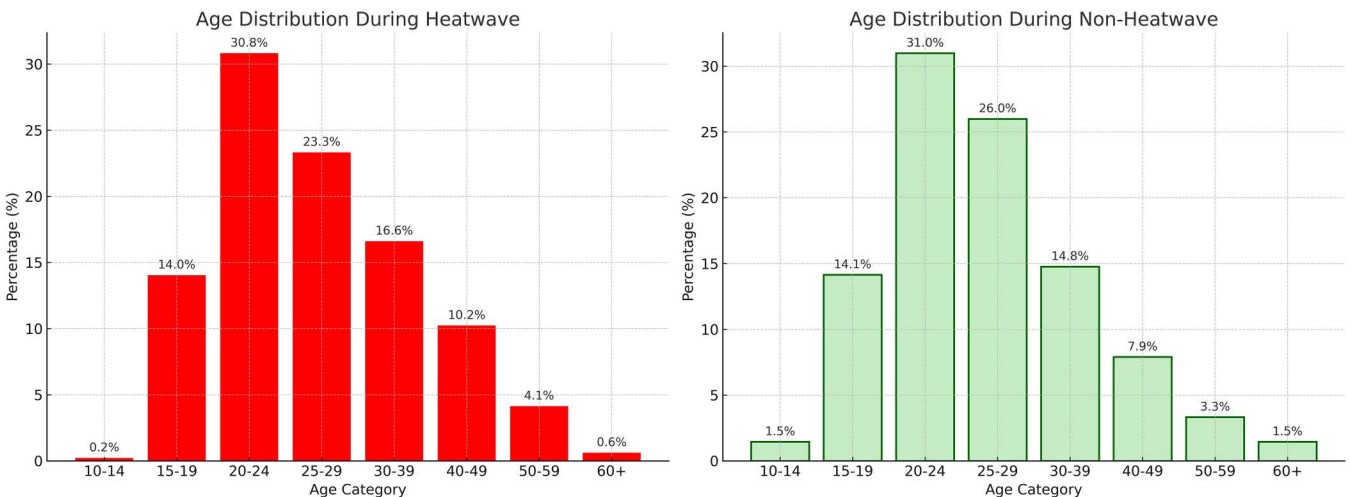

**Fig 3. Age Distribution during Heatwave vs. Non-heatwave periods.** Compares the age profiles of participants across both types of periods.

## Perceived Well-being During Heatwaves and Non-Heatwaves

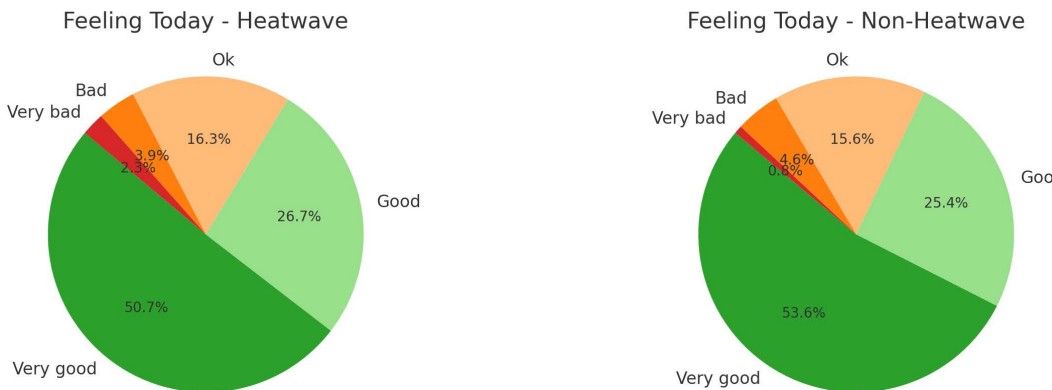

**Fig 4. Perceived well-being during heatwave and non-heatwave periods.** Presents how participants rated their general well-being at the time of the survey.

and stress were also more common (40.8% vs. 34.1%), while symptoms like diarrhea/vomiting and respiratory difficulties showed smaller differences between conditions (Fig 6). City-level patterns varied: in Port Harcourt (n = 325), heatwave-related symptoms were particularly high, with 53.5% reporting headaches, 33.3% skin irritation, and 45.2% low mood. Manila (n = 447) also showed elevated symptom rates during heatwaves, including itchy eyes (27.9%) and anxiety/stress (41.6%). In contrast, Dar es Salaam (n = 399) displayed more stable symptom reporting, with similar frequencies across both conditions. In contrast, the number of participants without symptoms was generally higher during non-heatwave periods (S5 Table).

Chi-square tests were used to examine the relationship between heatwave periods and reported well-being (feeling today and sleep quality) and health symptoms, both within individual cities and across all six cities combined. In Port Harcourt (n = 325), heatwave periods were significantly associated with both how participants felt that day and their sleep quality (p < 0.001). In Dar es Salaam (n = 399), only sleep quality showed a significant association (p = 0.002) (S4

## Perceived Sleep Quality During Heatwaves and Non-Heatwaves

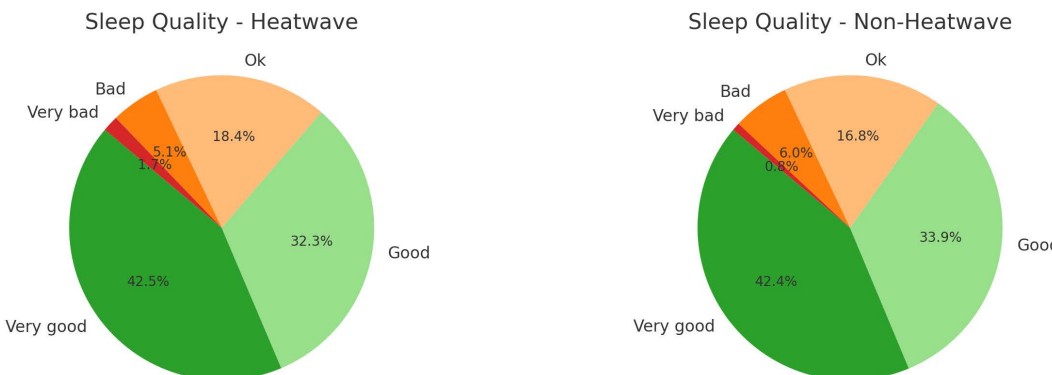

**Fig 5. Perceived sleep quality during heatwaves and non-heatwaves.** Participants' ratings of sleep quality from the previous night, shown by event type.

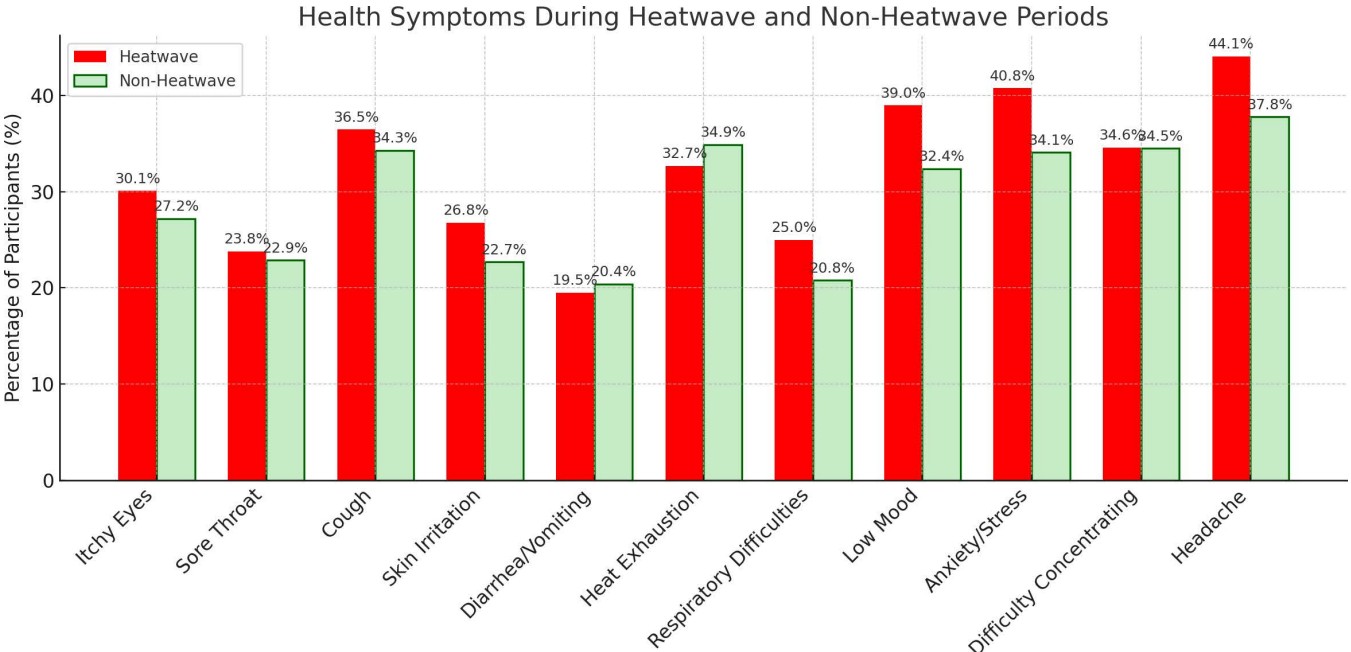

**Fig 6. Perceived health symptoms during heatwave and non-heatwave periods.** Participants reporting symptoms such as low mood, anxiety, and headaches during and after heatwaves.

Table). However, when data from all cities were pooled (n = 2269), no significant link was found between heatwaves and well-being.

Health symptoms showed variation by city. In Manila (n = 447), all reported symptoms were significantly associated with heatwave periods (p < 0.05). Ouagadougou (n = 244) showed a significant result for heat exhaustion only, while Kumasi (n = 293) showed no significant associations (S5 Table and S6 Table). In the pooled analysis, three symptoms, low mood (p = 0.008), anxiety or stress (p = 0.007), and headaches (p = 0.013), were statistically significant during heatwaves. These

were selected for further analysis using logistic regression to examine their independent associations with heatwave exposure.

The regression models, across cities, showed that participants during heatwave periods were significantly more likely to report symptoms of low mood, anxiety or stress, and headaches (Table 1). Compared to participants in non-heatwave periods, those during a heatwave periods were 1.38 times more likely to report low mood (p = 0.003), 1.36 times more likely to report anxiety or stress (p = 0.004), and 1.33 times more likely to report headaches (p = 0.007) (Table 1 and S6 Table).

Participants' younger age was significantly associated with higher odds of experiencing Low Mood, Anxiety or Stress, and Headache (OR = 0.97, *p* < 0.001). Regarding income, individuals earning $500–$1499 per month had increased odds of reporting Low Mood (OR = 1.97, *p* = 0.045) and Anxiety or Stress (OR = 2.90, *p* = 0.002), relative to those earning less than $100/month. Similarly, participants in the $100–$499 income group showed higher odds of Anxiety or Stress (OR = 1.64, *p* = 0.002) (Table 1).

Significant differences were also found between cities. Compared to Accra, participants in Dar es Salaam and Ouagadougou were significantly less likely to report low mood (OR = 0.56 and OR = 0.50) and anxiety/stress (OR = 0.57 and OR = 0.55). Participants in Port Harcourt were significantly more likely to report anxiety/stress (OR = 1.47, p = 0.007). For headaches, those in Dar es Salaam had significantly lower odds than those in Accra (OR = 0.74, p = 0.040). Although these city differences were statistically significant, the strength and direction of the associations varied.

## Disruptions to daily activities

Inactivity, i.e., 0 minutes of physical activity, was lower during heatwaves (14.5%) than during non-heatwave periods (16.8%). Meanwhile, a slightly higher proportion of participants reported engaging in more than 60 minutes of activity during heatwaves (20.4%) compared to non-heatwave days (19.8%). Port Harcourt (n = 325) recorded high levels of

**Table 1. Logistic regression results for health symptoms during heatwaves.**

| Outcome variables | Predictor variables | B | S.E. | Wald | df | Sig. | Exp(B) |
|---|---|---|---|---|---|---|---|
| Low Mood | Event (Heatwave) | 0.326 | 0.111 | 8.666 | 1 | 0.003 | 1.38 |
| | Age | -0.031 | 0.005 | 40.898 | 1 | <0.001 | 0.97 |
| | Income ($500-$1499) | 0.678 | 0.338 | 4.027 | 1 | 0.045 | 1.969 |
| | City (Dar es Salaam) | -0.579 | 0.155 | 13.97 | 1 | <0.001 | 0.56 |
| | City (Ouagadougou) | -0.701 | 0.174 | 16.238 | 1 | <0.001 | 0.496 |
| Anxiety/ Stress | Event (Heatwave) | 0.31 | 0.109 | 8.125 | 1 | 0.004 | 1.364 |
| | Age | -0.024 | 0.005 | 25.923 | 1 | <0.001 | 0.977 |
| | Income ($100-$499) | 0.497 | 0.159 | 9.738 | 1 | 0.002 | 1.644 |
| | Income ($500-$1499) | 1.064 | 0.337 | 9.991 | 1 | 0.002 | 2.897 |
| | Income (Don't Know) | 0.589 | 0.276 | 4.543 | 1 | 0.033 | 1.802 |
| | City (Dar es Salaam) | -0.568 | 0.154 | 13.622 | 1 | <0.001 | 0.567 |
| | City (Ouagadougou) | -0.602 | 0.172 | 12.240 | 1 | <0.001 | 0.548 |
| | City (Port Harcourt) | 0.387 | 0.144 | 7.253 | 1 | 0.007 | 1.472 |
| Headache | Event (Heatwave) | 0.288 | 0.107 | 7.2 | 1 | 0.007 | 1.333 |
| | Age | -0.032 | 0.005 | 47.581 | 1 | <0.001 | 0.968 |
| | City (Dar es Salaam) | -0.305 | 0.148 | 4.229 | 1 | 0.040 | 0.737 |

Adjusted odds ratios showing associations between heatwave and reported for low mood, anxiety/stress, and headaches controlling for age and income. Reference categories: Event = Non-heatwave; Income = less than $100/month; City = Accra. The table includes only predictors with p-values < 0.05. For full model results, see S6 Table.

extended activity in both periods, with 27.9% of participants engaging in more than an hour of physical activity during heatwaves and 30.8% during non-heatwave periods. Notably, no participants in Port Harcourt reported being completely inactive, i.e., zero minutes of physical activity in either event. In contrast, Dar es Salaam (n = 399) had the highest rate of inactivity during non-heatwave periods (23.7%) and lower levels of extended activity, with only 11.7% reaching more than 60 minutes during heatwaves (S7 Table). Overall, physical activity levels were slightly higher during heatwaves than in non-heatwave periods.

Disruptions to daily activities were common across all cities, with some variables showing notable differences between heatwave and non-heatwave periods (S8 Table). For instance, Port Harcourt (n = 325) experienced the most disruption during heatwaves across nearly all variables. Notably, 57.9% reported not having enough food and 57.1% reported needing more family assistance (S8 Table). Overall, the proportion of participants reporting not having enough food was higher during heatwaves (44.1%) compared to non-heatwave periods (37.6%), as was the reported need for additional family assistance (46.3% vs. 40.1%). In contrast, some disruptions, such as being late for school or work (33.7% vs. 34.3%) and missing school or work completely (24.6% vs. 24.3%), remained relatively stable across conditions. Access to water and healthcare appointments also showed only minimal differences (Fig 7).

Chi-square tests examined associations between heatwave periods and daily activity disruptions across cities. Several disruptions were significantly more common during heatwave periods. In Manila (n = 447), all variables, including lateness, missed school or work, missed healthcare, missed interviews, and changes to daily routines, were significantly associated with heatwave periods (p < 0.05) (S8 Table). In Accra (n = 561), significant associations were found for missed school or work, missed healthcare, lack of access to water, and increased need for family assistance. Dar es Salaam (n = 399) showed a significant association for missed school or work only (S8 Table).

Across all cities, only two disruptions, not having enough food and needing additional family support, were consistently significant (p < 0.05) and were selected for further analysis using logistic regression (Table 2 and S9 Table).

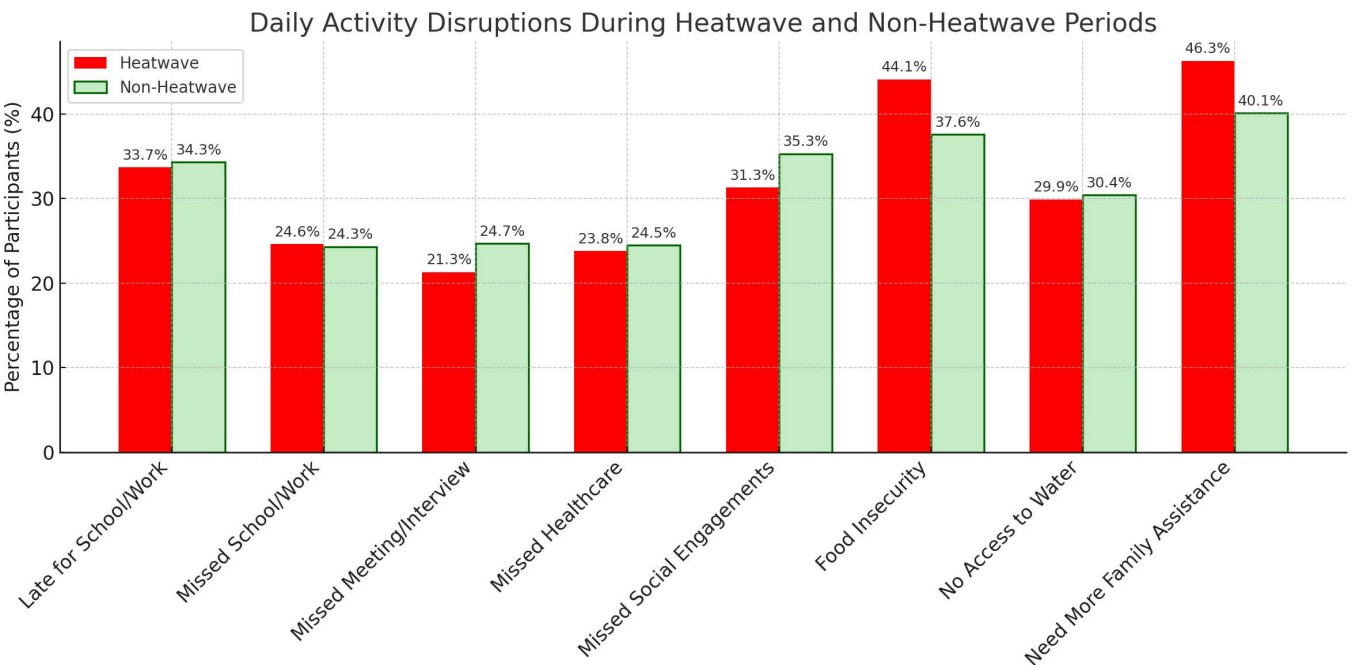

**Fig 7. Daily activity disruptions during heatwave and non-heatwave periods.** Reported daily disruptions such as missed work or school, and access to services.

The findings show that participants were 1.49 times more likely to report not having enough food (p = 0.009) and 1.59 times more likely to say they needed more assistance from family members (p = 0.002) when a heatwave occurred (Table 2).

Age was significantly associated with both outcomes. Younger participants were more likely to report not having enough food (OR = 0.956, p < 0.001) and a greater need for family support (OR = 0.96, p < 0.001). This group was 2.56 times more likely to report not enough food compared to women (p = 0.027). Income level showed statistically significant associations with disruptions to daily activities. Participants earning $100–$499 per month had significantly higher odds of experiencing Not Enough Food (OR = 1.43, p = 0.023) and Need More Family Assistance (OR = 1.39, p = 0.036). Those in the $500–$1499 income group also had significantly higher odds of Not Enough Food (OR = 1.87, p = 0.025) (Table 2).

City-level differences were significant for both outcomes. Compared to Accra, participants in Dar es Salaam and Manila were significantly less likely to report not having enough food during heatwaves (OR = 0.61 and OR = 0.72, respectively). In contrast, those in Port Harcourt were significantly more likely to report not having enough food (OR = 1.51, p = 0.004). For the outcome of needing more family assistance, participants in Manila were also less likely to report this need (OR = 0.76, p = 0.045), while those in Port Harcourt were significantly more likely to report needing family support during heatwaves (OR = 1.55, p = 0.002). (S9 Table for detailed analysis).

## Heatwave concerns and response satisfaction

Across all cities, the most frequently selected concern levels during heatwaves were 5 (31.8%) and 10 (22.1%), representing over half of all responses (Fig 8). In contrast, only 33.3% of participants chose these levels during non-heatwave periods. Nearly 30% reported no concern (0/10) when not experiencing a heatwave, highlighting a notable increase in concern with direct exposure. City-level patterns varied, but some, such as Accra and Manila, showed particularly high concern, with over 50% of participants selecting scores of 5 or higher. Others, like Dar es Salaam, had a larger share of respondents reporting no concern (53.2%) during non-heatwave conditions (S10 Table). Overall, most participants reported moderate to high levels of concern about heatwaves, particularly during periods of extreme heat.

Table 2. Logistic regression results for disruptions in daily activities during heatwaves.

| Outcome variables | Predictor variables | B | S.E. | Wald | df | Sig. | Exp(B) |
|---|---|---|---|---|---|---|---|
| Not Enough Food | Event(Heatwave) | 0.401 | 0.153 | 6.899 | 1 | 0.009 | 1.493 |
| | Age | -0.045 | 0.006 | 52.485 | 1 | 0 | 0.956 |
| | Gender (Other/Prefer Not to say) | 0.939 | 0.425 | 4.891 | 1 | 0.027 | 2.558 |
| | Income ($100-$499) | 0.355 | 0.156 | 5.168 | 1 | 0.023 | 1.426 |
| | Income ($500-$1499) | 0.627 | 0.279 | 5.043 | 1 | 0.025 | 1.871 |
| | Income (Don't Know) | -0.495 | 0.242 | 4.174 | 1 | 0.041 | 0.61 |
| | City(Dar es Salaam) | -0.493 | 0.149 | 10.863 | 1 | 0.001 | 0.611 |
| | City(Manila) | -0.329 | 0.138 | 5.693 | 1 | 0.017 | 0.719 |
| | City(Port Harcourt) | 0.409 | 0.143 | 8.138 | 1 | 0.004 | 1.505 |
| Need more family assistance | Event(Heatwave) | 0.464 | 0.151 | 9.517 | 1 | 0.002 | 1.591 |
| | Age | -0.041 | 0.006 | 46.039 | 1 | <0.001 | 0.96 |
| | Income ($100-$499) | 0.326 | 0.156 | 4.391 | 1 | 0.036 | 1.386 |
| | City(Manila) | -0.275 | 0.137 | 4.010 | 1 | 0.045 | 0.760 |
| | City(Port Harcourt) | 0.440 | 0.143 | 9.422 | 1 | 0.002 | 1.553 |

Adjusted odds ratios showing associations between heatwave and reported food insecurity and need for family support, controlling for age, gender, and income. Reference Category: Event = Non-heatwave; Gender = Female, Income = <$100. The table includes only predictors with p-values < 0.05. For full model results, see S9 Table.

Participants across all cities reported a range of satisfaction levels with heatwave responses. During heatwaves, the most common response was "Very Satisfied" (34.6%), followed by "Neutral" (33.3%), while 19.5% expressed dissatisfaction. In non-heatwave periods, satisfaction was even higher, with 48.6% of participants reporting being "Very Satisfied," and fewer choosing "Neutral" (26.0%) (Fig 9 and S11 Table). City-level patterns varied. In Port Harcourt (n = 325), 34.0% of participants were "Very Satisfied" during heatwaves, and only 11.0% reported any dissatisfaction. Dar es Salaam (n = 399) showed even stronger approval, with 72.0% selecting "Very Satisfied." Accra (n = 561), on the other hand, had a more even spread of responses, including 34.0% "Neutral," 26.3% "Very Satisfied," and 25.3% reporting dissatisfaction (S11 Table).

## Suggestions for healthier and sustainable cities

In response to the open-ended question, *'If you could do one thing to make your town/city healthier and more sustainable during heatwaves, what would it be?'*, the most frequent keywords from participants' emphasized priorities around environmental sustainability and public health. Common words included "clean", "trees," "plant,"" "recycling," "water," "sanitation," and "health." These reflect strong community interest in greener environments and improved basic services (Fig 10).

A thematic analysis of participant responses identified five key priorities for healthier urban living: Urban Greening, Water and Sanitation, Clean and Healthy Environment, Health Education and Services, and Civic Engagement. These themes, illustrated in Fig 11 and expanded in (S12 Table), reflect actionable ideas rooted in community experiences and aspirations.

Participants' responses to the open-ended question revealed five main themes. Urban greening was the most common, with suggestions to expand green spaces and improve air quality, such as "plant trees for shade during extreme heat" and "encourage people to plant one tree for every household." Under water and sanitation, participants called for better infrastructure and cooling features, including a "continuous flow of water through our taps" and public "fountains or ponds to

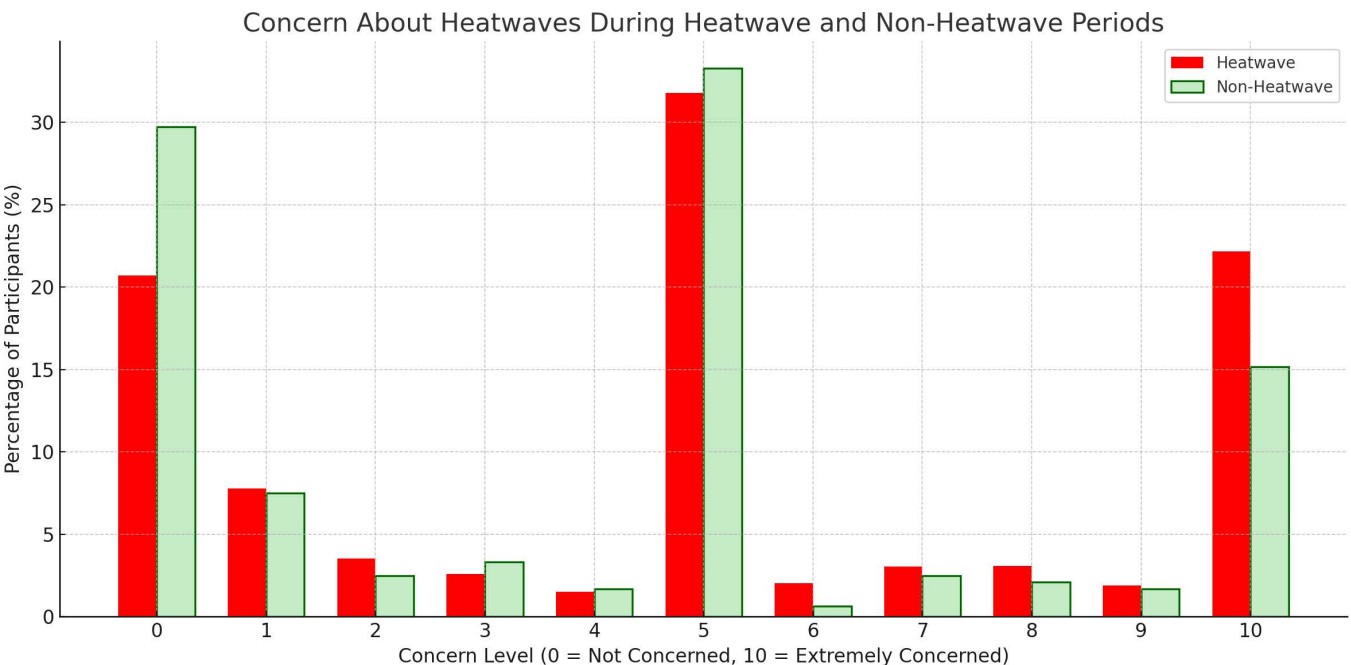

**Fig 8. Concern about heatwaves during heatwave and non-heatwave periods.** Participants' concern about heatwaves, rated on a scale from 0 (no concern) to 10 (very concerned), by event type.

Global Public Health

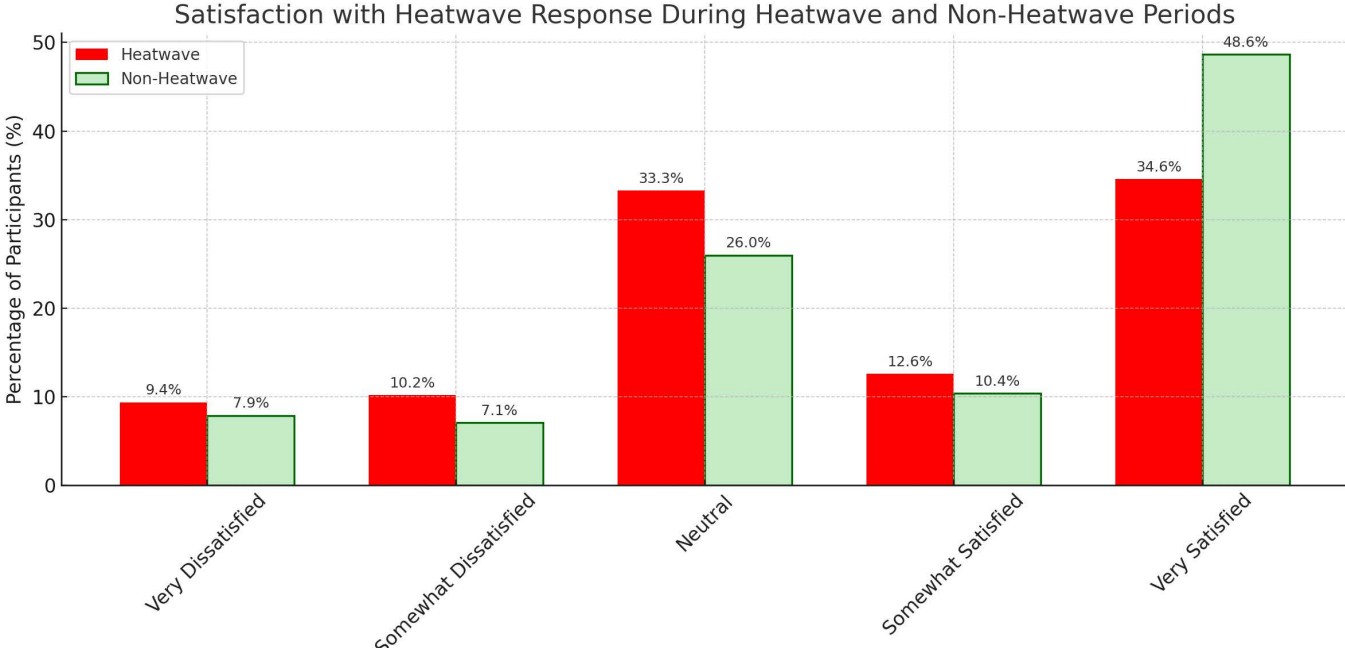

**Fig 9. Satisfaction with heatwave response during heatwave and non-heatwave period.** Participants' views on how well their city responded to recent heatwaves, based on satisfaction level.

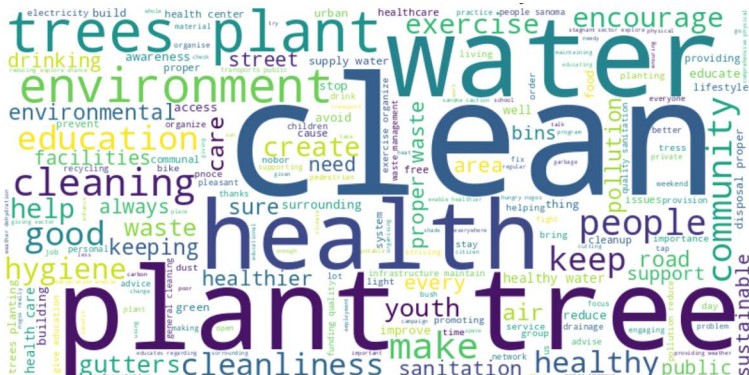

**Fig 10. Word cloud of the most frequent keywords from participants' suggestions for healthier and sustainable cities.** Word cloud showing the most frequent suggestions from participants on how to improve their cities during heatwaves.

cool the air." The theme of a clean and healthy environment focused on waste management and hygiene, with recommendations like "provide bins on the streets to prevent pollution" and keep areas "free of stagnant water and bushy grass." Health education and services emphasized awareness and local care, with calls to "create more awareness on health and environmental issues" and "build a well-equipped health center." Lastly, civic engagement highlighted the role of young people in city upkeep, suggesting efforts to "mobilize the youth to clean the community" and "create sustainable jobs and education for the youth."

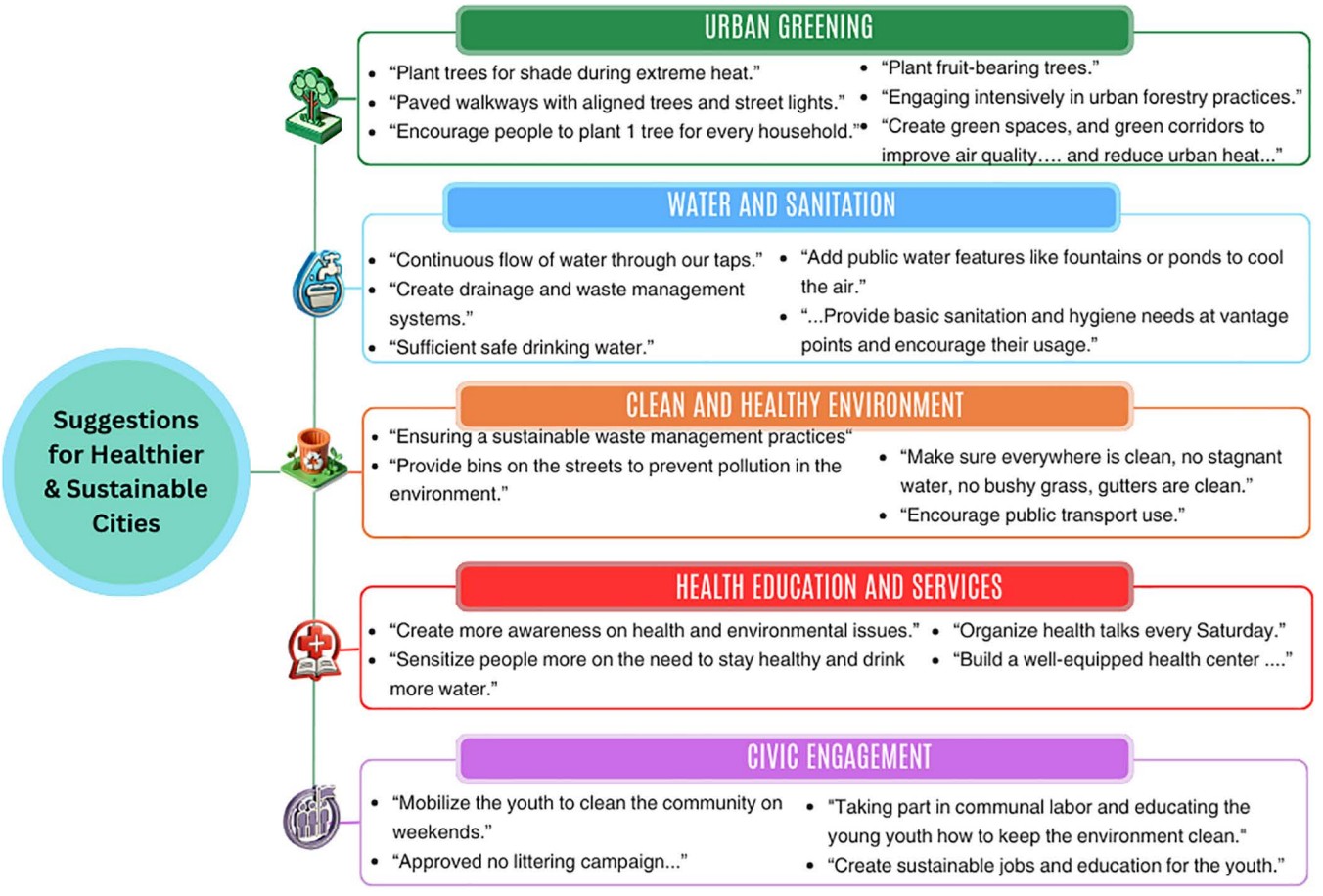

**Fig 11. Themes, and illustrative quotes from participant responses on making cities healthier and more sustainable.** Key themes from participants' open-ended responses, i.e., urban greening, water and sanitation, clean environments, health education and services, and civic engagement, along with selected example quotes from youth and caregivers.

## Discussion

### Summary of findings

Our findings indicate that heatwaves are associated with a higher prevalence of self-reported health symptoms such as low mood, anxiety/stress, and headaches. For example, participants in Port Harcourt and Manila reported higher rates of symptoms during heatwaves, while Dar es Salaam showed less variation between heatwave and non-heatwave periods.

Additionally, heatwaves appear to exacerbate disruptions in daily activities, particularly not enough food and the need for additional family support. These disruptions were most pronounced in Port Harcourt, where over half of participants reported not having enough food and relying more on family support during heatwaves. By contrast, other cities reported lower but still notable increases in these disruptions.

Beyond quantifiable impacts, participants expressed insights into how young people and caregivers imagine healthier, more resilient urban futures. Through thematic analysis of open-ended responses, five core priorities emerged: urban greening, improved sanitation, cleaner environments, better health education and services, and civic engagement. While these themes were common across all locations, some differences emerged. For instance, respondents in Accra

and Kumasi emphasized sanitation and water, while those in Manila highlighted environmental cleanliness and waste management.

## Interpretations of results

The impacts of heatwaves across the six cities were not evenly distributed. Younger participants and those with lower incomes reported more health symptoms and greater disruptions to daily life (Tables 1 and 2). These findings align with research showing that CYP are physiologically more vulnerable to heatwaves due to their more limited ability to regulate body temperature [1,30], and their heightened susceptibility to psychosocial stressors. People with fewer resources are also less likely to have access to cooling, healthcare, or flexible work arrangements, all of which can protect against heat-related effects [15,53,54].

City-level differences also emerged. Compared to Accra, participants in Dar es Salaam and Ouagadougou reported lower odds of reporting low mood and anxiety/stress. This may be due to a combination of intense heat, limited green space, crowded living conditions, and weak social support systems [36,40,46,55]. These differences highlight the need for localised interventions that account for the distinct environmental and socioeconomic conditions shaping residents' vulnerability to heat-related stress.

Health symptoms particularly mental health were consistently reported during heatwaves across all cities. Participants described more low mood, anxiety, stress, and headaches, contributing to the growing evidence that extreme heat is linked to deteriorating mental health distress [10,56,57]. Possible explanations include changes in brain chemistry and worsening of existing mental health conditions [56,57]. Headaches, in particular, may reflect physiological stress responses such as dehydration, thermal strain, or disrupted sleep patterns, all of which are exacerbated during periods of extreme heat [40,55,58]

At the same time, some studies introduce nuance and resilience. Vergunst and Berry (2022) noted that children may develop resilience to climate stressors [10]. Similarly, Cuartas and Vergunst (2025) argue that nurturing care and strong social relationships can buffer the psychological effects of environmental stressors [13]. Household-level adaptations, such as adjusting routines, using fans or air conditioners, and improving ventilation or shading, may also mitigate heat-related distress [45]. These responses could explain why certain variables, like general well-being and sleep quality, did not show consistent change between heatwave and non-heatwave periods in this study data, despite other studies showing strong declines in sleep quality and general well-being during heatwaves [24,25,59].

Physical activity levels slightly increased during heatwaves. This is consistent with findings from Ravanelli et al. (2023), who reported that some CYP remain physically active even during heatwaves, highlighting variability in behaviour, awareness, and environmental conditions [60]. Similarly, Morrison et al. (2022) observed that CYP with greater aerobic fitness demonstrated improved heat tolerance and fewer behavioural changes in response to high temperatures [61]. However, this contrasts other studies, particularly from high-income, service-based economies, where CYP tend to reduce physical activity during extreme heat due to increased access to climate-controlled environments and greater behavioural flexibility [12,62,63]. For example, preschool-aged children have been shown to walk less, take fewer steps, and exhibit lower peak heart rates on hotter days. Access to shaded play areas has been associated with sustained activity and improved thermal comfort [64].

In addition to physical and mental health challenges, participants reported that not having enough food and the need for family support increased during heatwaves, particularly among younger, lower-income and participants in Port Harcourt. These outcomes are likely linked to the additional strain extreme heat places on daily life, reducing income-earning capacity, disrupting food storage and preparation, and increasing caregiving needs [1,53,65]. Younger individuals may be more affected due to their greater reliance on caregivers, while low-income households face greater difficulty absorbing the financial and logistical impacts of heatwaves to access food and provide family support [1,13].

These findings are consistent with studies showing that heatwaves can intensify economic and social vulnerabilities in cities, especially where public infrastructure and services are limited [1,13,53,65]. Specifically, Kroeger (2023)

demonstrated that extreme heat is associated with short-term increases in household food insecurity across multiple countries, primarily mediated through income effects, closely aligning with the patterns observed in this study [65]. Similarly, Hossain et al. (2024) highlighted the role of socioeconomic inequality in shaping urban households' capacity to adapt to heat, reinforcing the heightened vulnerability of low-income groups [53]. The increased need for family support, especially among younger participants, also aligned with Cuartas and Vergunst's (2025) argument that children are particularly exposed to climate-related stressors due to their dependence on caregivers [13]. However, these findings should be interpreted alongside studies that offer nuance. Amankwaa and Ampomah (2025) further note that many households actively use behavioural adaptations, such as altering cooking schedules, improving ventilation, and modifying routines, to manage heat impacts, which may mitigate some of the food- and care-related stress reported here [45].

These community-derived themes closely align with global frameworks like the WHO Healthy Cities initiative and UNICEF's climate resilience priorities for children [66–68], reinforcing the legitimacy and relevance of youth-informed planning in urban adaptation.

The perception of satisfaction with city responses should be interpreted with caution. In some contexts, this may reflect psychological resilience or adaptive preferences; in others, it could be a sign of normalized deprivation, where poor infrastructure is expected rather than contested [53]. Perceived satisfaction with city interventions may be partly attributable to visible mitigation efforts, such as emergency alerts, cooling centers, and other adaptive infrastructure [62]. In many cities, acceptance of city responses may reflect low expectations shaped by ongoing exposure to heat stress in dense, poorly serviced environments, where limited vegetation, weak ventilation, and insufficient public infrastructure leave household to rely on their own adaptative measures [53].

Perceptions of heatwave responses may also be influenced by whether events were formally recognized by local authorities. In our study, only Manila and Accra had nationally recognized heatwaves accompanied by public alerts, which may have heightened awareness and shaped expectations of city action. By contrast, in other study cities where events were not officially designated as heatwaves, participants may have perceived them as part of ongoing seasonal heat. This difference in recognition could partly explain variation in reported satisfaction and concern, highlighting the importance of both formal acknowledgment and effective communication in shaping public perceptions of extreme heat.4; Lines.

## Strengths of the study

This study included participants from six diverse cities, which allowed us to compare experiences of CYP across different urban environments. The use of real-time temperature data and an automated system to identify heatwave periods helped ensure that we captured event periods consistently and accurately. The sample size was large and included a wide range of ages, income levels, which adds depth to the findings. By combining survey data with open-ended responses, the study not only measured the impact of heatwaves but also gave space for participants to share their ideas for healthier cities. The involvement of a Youth Advisory Group in designing the survey and shaping the analysis added further value and ensured the research remained grounded in the experiences of young people.

## Implications for policy and practice

The findings highlight the need for climate-sensitive urban planning that reflects the perspectives of CYP. While participants stressed the importance of green spaces, policies could also consider blue infrastructure, balancing its cooling potential with possible health risks. Interventions could focus on expanding green and blue infrastructure, improving access to safe water, food, and sanitation, and providing stronger support for families during climate shocks. Careful planning of blue infrastructure is necessary, as water bodies may, in some cases, increase humidity or limit night-time cooling, and can also create conditions for mosquito-borne and water-related diseases. In addition, surface colour and material composition strongly influence thermal comfort and should be incorporated into design strategies. Finally, the willingness of young people to engage in civic action highlights their potential role in building more resilient cities through participatory governance.

 

Urban heatwave preparedness must move beyond physical infrastructure to include social protection, mental health support, and climate education tailored to young urban populations. These findings align with existing global frameworks, including the WHO Healthy Cities initiative and UNICEF's climate resilience priorities for children [66,67], reinforcing the legitimacy and relevance of youth-informed planning in urban climate adaptation. For policy-makers, this alignment strengthens the case for funding mechanisms that specifically support youth-led or community-driven adaptation efforts, ensuring that investments reflect the lived realities and priorities of those most affected by climate risks.

At the same time, effective policies must be adapted to the socio-cultural and climatic contexts of individual cities. Rather than limiting the value of these results, this heterogeneity points to the importance of locally grounded strategies. Complementary participatory and systems-oriented approaches can build on these insights, ensuring that policies are both evidence-based and context-specific.

### Recommendations for future research

Future research should consider longitudinal designs that follow the same participants over time to better assess whether changes in reported health outcomes or daily disruptions are linked to heatwave exposure or differences in sample composition. Combining online recruitment with offline or community-based approaches would improve inclusion of individuals with limited internet access, who may be more vulnerable to extreme heat. Collecting data across multiple heatwave and non-heatwave periods could help increase sample sizes and enable more balanced comparisons across cities. Additionally, detailed city-specific analyses, along with an examination of within-city differences, such as demographics, social-cultural, or access to services, would provide a clearer understanding of how heatwaves affect health, wellbeing, and daily life. Furthermore, integrating tools such as wearable devices, mobile apps, or local temperature sensors could improve exposure measurement and enhance the accuracy of findings. Finally, given the complexity of urban systems, future research should also complement survey-based approaches with systems-oriented methods (e.g., spatial modelling, participatory systems mapping) to better capture the interconnections between infrastructure, social dynamics, and climate exposures.

### Conclusion

This study shows the uneven impacts of heatwaves on CYP across diverse urban contexts, shaped by age, socio-economic status, and location. Heatwaves were associated with heightened health symptoms, daily disruptions, and increased risks such as food insecurity, while participants also identified clear priorities for strengthening city resilience.

The results highlight CYP-informed and context-specific adaptation strategies that reflect the lived realities of CYP in rapidly urbanising settings. Alongside investments in green and blue infrastructure, attention to food and water systems, accessible health care, and family support during climate shocks will be essential for healthier and more sustainable cities. Participatory governance can further ensure these strategies are equitable and climate-resilient.

By centering the voices of CYP in the Global South, this study contributes critical evidence to the climate and health field and highlights the need for urgent, locally grounded action to address the rising risks of extreme heat.

### Supporting information

**S1 File. Protocol for selecting the CCC Action Lab focal cities.** Describes the criteria and process used to identify and select the 178 focal cities used for the event identification in the study.
(DOCX)

**S2 File. Advertisements for the six cities.** Includes screenshots and copies of the online recruitment advertisements used in each of the six study cities.
(DOCX)

**S3 File. Data collection tools.** Presents the questionnaires in 12 languages used in the study.
(DOCX)

**S4 File. Quantitative Dataset.** Contains the anonymized, cleaned dataset used for quantitative analysis in this study.
(DOCX)

**S5 File. Data analysis plan.** Outlines the planned analytical approach for both quantitative and qualitative data.
(DOCX)

**S6 File. Sample mayor's email.** Provides an example of the outreach email used to inform city officials of the data collection in their city.
(DOCX)

**S1 Fig. Event sampling flowchart.** Illustrates the process for identifying of eligible heatwave and non-heatwave periods and participant recruitment in the six cities.
(DOCX)

**S1 Table. Sample size calculation per city.** Displays the estimated minimum number of responses required per city for robust analysis.
(XLSX)

**S2 Table. Advertisement performance and survey engagement.** Summarizes key metrics on ad impressions, click-throughs, and completed surveys by city.
(DOCX)

**S3 Table. Demographic characteristics.** Provides a breakdown of age, gender, and parental status for survey respondents.
(DOCX)

**S4 Table. Well-being (feeling today and sleep quality).** Presents participant-reported measures of general well-being and sleep quality.
(DOCX)

**S5 Table. Health symptoms.** Lists the self-reported physical and emotional health symptoms during heatwave and non-heatwave periods.
(DOCX)

**S6 Table. Logistic regression results for health symptoms during heatwaves.** Shows adjusted odds ratios for the likelihood of experiencing health symptoms during heatwave periods.
(XLSX)

**S7 Table. Physical activity.** Summarizes reported levels and changes in physical activity during extreme heat periods.
(DOCX)

**S8 Table. Disruptions in daily activities.** Details interruptions to school, work, and caregiving activities during heatwaves.
(DOCX)

**S9 Table. Logistic regression results for daily activity disruptions during heatwaves.** Presents adjusted models assessing the association between heatwaves and daily activity disruptions.
(XLSX)

**S10 Table. Concern about heatwaves.** Displays participants' levels of concern about current and future heatwaves.
(DOCX)

**S11 Table. Satisfaction with heatwave response.** Summarizes Children, Young People and Parents perspectives on their satisfaction on how well their cities responded to recent heatwaves.
(DOCX)

**S12 Table. Thematic codebook and coded qualitative dataset.** Provides the codebook used for qualitative analysis and an excerpt of coded text illustrating key themes.
(XLSX)

## Acknowledgments

We are sincerely grateful to the CYP and parents who took part in the real-time survey activities. We also extend our appreciation to the local teams and partner organizations across the participating cities for their role in facilitating data collection. Special thanks to the Empower Agency for their valuable contribution to the recruitment process and for providing training to staff involved in online data collection.

## Author contributions

**Conceptualization:** Constance Bwire, Rachel Juel.

**Data curation:** Constance Bwire.

**Formal analysis:** Constance Bwire, Ana Bonell.

**Funding acquisition:** Robert Hughes.

**Investigation:** Constance Bwire.

**Methodology:** Constance Bwire.

**Project administration:** Constance Bwire.

**Visualization:** Constance Bwire.

**Writing – original draft:** Constance Bwire.

**Writing – review & editing:** Constance Bwire, Rachel Juel, James Milner, Gabrielle Bonnet, Ana Bonell, Shunmay Yeung, Harshita Umesh, Roberto Picetti, Sudheer Kumaar Kuppili, Jessica Gerard.

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
