## [Decision Letter · Decision Letter 0]

26 Aug 2025

PGPH-D-25-01526

Living through the heat: How urban children and young people experience and envision healthier cities

Dear Dr. Bwire,

Thank you for submitting your manuscript to PLOS Global Public Health. After careful consideration, we feel that it has merit but does not fully meet PLOS Global Public Health’s publication criteria as it currently stands. Therefore, we invite you to submit a revised version of the manuscript that addresses the points raised during the review process.

We look forward to receiving your revised manuscript.

Kind regards,

Dinesh Bhandari, Ph.D.

Academic Editor

Journal Requirements:

Additional Editor Comments (if provided):

Reviewers' comments:

Reviewer's Responses to Questions

**Comments to the Author**

1. Does this manuscript meet PLOS Global Public Health’s publication criteria?

Reviewer #1: Yes

Reviewer #2: Partly

2. Has the statistical analysis been performed appropriately and rigorously?

Reviewer #1: Yes

Reviewer #2: Yes

3. Have the authors made all data underlying the findings in their manuscript fully available (please refer to the Data Availability Statement at the start of the manuscript PDF file)?

Reviewer #1: Yes

Reviewer #2: No

4. Is the manuscript presented in an intelligible fashion and written in standard English?

Reviewer #1: Yes

Reviewer #2: Yes

Reviewer #1: Overall, an insightful and clearly structured paper on a pressing topic related to extreme heat events in cities of the Global South and their impact on children and young people. Here are only a few minor suggestions for consideration. First, the study failed to address ethics approval. A sentence would be useful to highlight which institution granted permission. In terms of the selection of cities, it is worth highlighting that urban form, density, and size were not considered. As the heat island effects of cities due to the number of hard surfaces are usually higher than those of the surrounding area, it needs to be made clear why this was not considered. Additionally, regarding engagement methods, it would be beneficial to elaborate further on why a survey was chosen as the preferred method over others tailored to the user group. It is briefly mentioned in the limitations, but it deserves more attention, as there is substantial evidence in the area of participation with and for children and young people that other methods allow for much richer and nuanced insights into environmental perception. Furthermore, the study has significant limitations due to the complexity of urban systems and the existence of more appropriate methods to investigate the nexus through systems approaches. Care must be taken around the conclusion on sustainable and healthy cities; some cross connections deserve more clarity, especially given that food system design (I assume this includes drinking as well) is one important factor in the data findings. Yet this was not addressed in the suggestions. The paper mentions green infrastructure, yet despite mentioning water, you might also wish to include blue infrastructure. However, there are studies on bioclimate that found that water bodies, under particular circumstances, can amplify heat effects and impact human health in other ways. Tipping ecology, including more mosquito- and vector-borne diseases, etc. Surface colours and material composition determine human comfort in different parts of those cities too. All of the selected cities are located in the Global South and contain slums or informal settlements in which the urban poor reside. While the study acknowledges limitations in terms of SES, given the sample size, it might be worth focusing on the responses of participants from such communities. Perhaps this might be worth future research in combination with different methods. Hence, I suggest reevaluating the study's strengths and limitations more critically from an urban and participatory research perspective on vulnerable population groups, as the limitations outweigh the benefits of the method in relation to the theme of inquiry. Equally, I suggest caution around the recommendation for policy, but highlight that other methods might achieve more meaningful and nuanced outcomes and impact. Socio-cultural aspects of place indicate that each of those cities is quite heterogeneous, and bio-climatic conditions can vary from place to place.

Reviewer #2: Dear Authors,

This is a very important topic and my comments are meant to strengthen the manuscript.

1. the statement in the introduction "we know little about how [children] experience their changing urban environments, especially during extreme event" is not strictly true. Please see these refrences:

Salima Meherali, Yared Asmare Aynalem, Saba Un Nisa, Megan Kennedy, Bukola Salami, Samuel Adjorlolo, Parveen Ali, Kênia Lara Silva, Lydia Aziato, Solina Richter, Zohra S Lassi - Impact of climate change on child outcomes: an evidence gap map review: BMJ Paediatrics Open 2024;8:e002592.

https://futureofchildren.princeton.edu/sites/g/files/toruqf2411/files/media/children_and_climate_change_26_1_full_journal.pdf

https://www.austlii.edu.au/au/journals/AUJlEmMgmt/2020/26.pdf

2. Please provide a citation for this statement: "A child born today risks living in a world up to four degrees Celsius warmer than the pre-industrial average..."

3. The paper misses out on discussing the literarure on the link between heat waves and heat stress. And more specifically, on heat stress both during and after heatwaves, which is driven by a complex interplay of climatic, infrastructural, and socio-environmental factors. These elements not only intensify the immediate thermal burden but also prolong discomfort and health risks well beyond the peak of a heatwave. Because this inadequate problem identification, I feel the paper lacks depth and understanding of what heat stress is and how that impacts health and well-being. For example, there is no discussion on how cities suffer more due to High Ambient and Surface Temperatures, Urban Heat Island (UHI) Effect, Relative Humidity, Reduced Wind Flow and Sky Visibility, Limited Green and Blue Infrastructure etc. Most importantly,

Heat stress is not restricted to heat waves. This is perhaps critical in explaining some of the study findings that found no difference in sleep patterns or physical activity levels during or after the heat waves. Right now the study lacks explanatory power due to inadequate context setting.

4. The limitations of the study should be discussed in the methods section and not at the end. I really missed any discussion of whether the periods chosen as heat waves are perceived as such in the study cities. Were these recognized as "heatwaves" by the local governments? Because that has a psychological impact in people's perception of heatwaves.

5. A major weakess of the design is that the study did not follow the same participants over time, making it difficult to assess whether changes in outcomes were directly linked to heatwave exposure. The study may have underestimated the true impact of heatwaves on the most vulnerable populations due to the exclusion of those without internet access or those less likely to participate in online surveys. Moreover, sample sizes varied across cities, with smaller and uneven samples during non-heatwave periods, potentially influencing the findings and limiting the generalizability of results.

6. For readers, it would be good to have some background information on the cities chosen, their level of urbanization, development and climate burdens.

7. Overall, I feel, this paper needs a major repositioning with greater engagement with literature and better discuss of the results.

**Do you want your identity to be public for this peer review?** For information about this choice, including consent withdrawal, please see our Privacy Policy

Reviewer #1: **Yes: ** Greg Mews

Reviewer #2: No

---

## [Editor Report · Decision Letter 1]

24 Sep 2025

Living through the heat: How urban children and young people experience and envision healthier cities

PGPH-D-25-01526R1

Dear Dr Bwire,

We are pleased to inform you that your manuscript 'Living through the heat: How urban children and young people experience and envision healthier cities' has been provisionally accepted for publication in PLOS Global Public Health.

Best regards,

Dinesh Bhandari, Ph.D.

Academic Editor
